# Prevalence of *Vibrio parahaemolyticus* Causing Acute Hepatopancreatic Necrosis Disease of Shrimp in Shrimp, Molluscan Shellfish and Water Samples in the Mekong Delta, Vietnam

**DOI:** 10.3390/biology9100312

**Published:** 2020-09-25

**Authors:** Tran Thi Hong To, Haruka Yanagawa, Nguyen Khanh Thuan, Du Minh Hiep, Doan Van Cuong, Ly Thi Lien Khai, Takahide Taniguchi, Ryoichi Kubo, Hideki Hayashidani

**Affiliations:** 1Division of Animal Life Science, Institute of Agriculture, Tokyo University of Agriculture and Technology, 3-5-8 Saiwai-cho, Fuchu-shi, Tokyo 183-8509, Japan; tthongto@tvu.edu.vn (T.T.H.T.); haruka330927@gmail.com (H.Y.); nkthuan50@gmail.com (N.K.T.); taniguti@cc.tuat.ac.jp (T.T.); 2School of Agriculture and Aquaculture, Tra Vinh University, 126, Hamlet 4, Ward 5, Tra Vinh, Vietnam; hiepminhcp84@gmail.com; 3Faculty of Agriculture, Can Tho University, Campus II, 3/2 Street, Ninh Kieu District, Can Tho City, Vietnam; ltlkhai@ctu.edu.vn; 4Research Institute for Aquaculture No.2, 116, Ward Da kao, District 1, Ho Chi Minh, Vietnam; vancuongdisaqua@gmail.com; 5Kanto Chemical Co. Inc., 2-1, Nihonbashi Muromachi 2-chome, Chuo-ku, Tokyo 103-0022, Japan; kubor@gms.kanto.co.jp

**Keywords:** antimicrobial resistance, *pir^vp^* genes, serotype, *Vibrio parahaemolyticus*

## Abstract

**Simple Summary:**

Recently, *Vibrio parahaemolyticus* has been identified as an important agent of acute hepatopancreatic necrosis disease in shrimp. In Vietnam, this disease has appeared since 2010 and caused a big economic loss for shrimp farming. However, the information of this agent in Vietnam has been not fully understood. This study aims to investigate the prevalence of shrimp pathogenic *Vibrio parahaemolyticus* and several it’s characteristics in the Mekong Delta of Vietnam. A total of 481 shrimp and molluscan shellfish samples from retail shops and farms and 64 water samples from shrimp and molluscan shellfish farms were examined for the presence of pathogenic *strains. The p*athogenic strains were isolated in 0.7% of molluscan shellfish samples from retail shops, 9.9% of shrimp samples from shrimp ponds, and 4.8% of water samples from shrimp ponds. These strains were classified into two types of O antigen (O1 and O3), in which O1 was the predominant. They showed resistance to several antimicrobial agents, multidrug resistance and pathogenicity to experimental shrimp. These results indicate that shrimp pathogenic *Vibrio parahaemolyticus* is widely prevalent in environment in the Mekong Delta, Vietnam. These findings can be used for understanding the risk of shrimp pathogenic *Vibrio parahaemolyticus* in the Mekong Delta.

**Abstract:**

A total of 481 samples, including 417 shrimp and molluscan shellfish samples from retail shops and farms and 64 water samples from shrimp and molluscan shellfish farms in the Mekong Delta located the southern part of Vietnam, were examined for the presence of *Vibrio parahaemolyticus* (*Vp*_AHPND_) caused acute haepatopancreatic necrosic disease (AHPND) in shrimp. *Vp*_AHPND_ strains were isolated in two of 298 (0.7%) molluscan shellfish samples from retail shops, seven of 71 (9.9%) shrimp samples from shrimp ponds, and two of 42 (4.8%) water samples from shrimp ponds. *Vp*_AHPND_ strains were classified into two types of O antigen, including O1 and O3, in which O1 was the predominant. *Vp*_AHPND_ strains isolated showed high resistance rates to colistin (100%), ampicillin (93.8%), and streptomycin (87.5%). These results indicate that *Vp*_AHPND_ is widely prevalent in environment in the Mekong Delta, Vietnam.

## 1. Introduction

Acute hepatopancreatic necrosis disease (AHPND), formally called early mortality syndrome (EMS), in shrimp was firstly detected in China in 2009 and subsequently reported in some Southeast Asian countries [1], Mexico [2,3], South America [4], and USA [5]. The disease might occur within 20–30 days after post- larvae released in ponds and causes a mortality of up to 100% [6]. Typically, shrimp infected AHPND show empty stomach and midgut, pale to white of hepatopancreas (HP) and atrophy of HP with the dysfunction of R-, B-, F-, and E- cells [1]. Loc et al. [7] indicated the specific strain of Vibrio parahaemolyticus (*Vp*_AHPND_) was the causative agent of AHPND. The following study conducted by Lee et al. [8] found the Photorhabdus insect-related (*pir*) toxin like genes (*pirAB^vp^*) encoded on plasmid of *V. parahaemolyticus* played as virulent genes responding to this disease in shrimp.

In Vietnam, AHPND has been announced since 2010 and caused a massive loss of shrimp farming in this country, particularly in the Mekong Delta in the South region, which provides approximately 95% of total shrimp production in the country [1,9]. The shrimp disease referred as AHPND spread on approximately 39,000 ha of shrimp farm in the Mekong Delta [6] and caused a loss at about 50% of stocking shrimp at the early stage in this area in 2012 [10]. Although V. parahaemolyticus was identified as the causative agent of AHPND in shrimp in Vietnam [7], up to now the prevalence of this pathogenic bacteria in environment in the Mekong Delta has been not fully understood. This research aimed to isolate *Vp*_AHPND_ in the Mekong Delta of Vietnam and examine some characteristics of this pathogen, such as pathogenicity, serotype, and antimicrobial resistance.

## 2. Materials and Methods

### 2.1. Sample Collection

A total of 481 samples, including 417 shrimp and molluscan shellfish samples from retail shops and farms and 64 water samples from shrimp and molluscan shellfish farms in the Mekong Delta, were collected in this study. The retail samples consisting of 32 shrimp samples (banana shrimp and greasyback shrimp) and 298 molluscan shellfish samples (white hard clam, blood cockle, mud clam, harf crenate ark, and antique ark) were purchased from wet markets in Can Tho city and Tra Vinh province in 2015 and 2016. The farming samples, including 71 white leg and black tiger shrimp samples and 16 white hard clam samples, were collected from 71 intensive shrimp ponds and two white hard clam farms, respectively, in Tra Vinh province in 2016 and 2017. The water samples, including 42 and 22 samples, were collected from 42 intensive shrimp ponds and two white hard clam farms, respectively, in Tra Vinh province in 2016.

### 2.2. Isolation and Identification of V. parahaemolyticus

For shrimp and molluscan shellfish samples, 25 g of each sample was mixed with 225 mL of alkaline peptone water (APW) (Nissui Co., Tokyo, Japan) in sterile stomacher bag to form homogenate solution. For water samples, 100 mL of each sample was mixed with 100 mL of two times high concentrate APW. All of the samples were incubated at 37 °C for 18 h. After that, a loopful of enrichment culture was inoculated on thiosulfate-citrate-bile salts-sucrose (TCBS) agar (Nissui Co., Tokyo, Japan) and CHROMagar Vibrio (CV) agar (CHROMagar, Paris, France) and incubated at 37 °C for 18 h. After incubation, green colonies on TCBS agar and mauve colonies on CV agar were picked up and subjected to biochemical tests. The strains showing glucose fermentation without gas production, lysine-positive, mobility-positive, indole-positive, oxidase-positive, VP- negative, no growth without NaCl, and growth from 3 to 8% NaCl were identified as *V. parahaemolyticus*. *V.*
*parahaemolyticus* strains identified was confirmed by PCR targeting the species-specific *toxR* gene, as described by Kim et al. (1999) [11].

### 2.3. Detection of Pathogenic Genes

The DNA of *V.*
*parahaemolyticus* strains was extracted using the boiling method. A loopful of colonies on the nutrient agar (NA) (Nissui Co., Tokyo, Japan) plate was mixed with 1 mL of sterile deionized distilled water into eppendorf tubes. Subsequently, the mixture was boiled at 100 °C for 10 min and centrifuged at 10,000 rpm for 10 min. After that, 500 µL of supernatant was moved into new eppendorf tube and kept at −20 °C for further use. The *pirAB^vp^* genes were examined using multiplex PCR following the protocol described by Han et al. (2015) [12]. The conditions of multiplex PCR for *pirAB^vp^* gene amplification were set at one cycle of 94 °C for 3 min., followed by 35 cycles of amplification consisting of denaturation at 94 °C for 30 s, annealing at 60 °C for 30 s, and extension at 72 °C for 30 s, and then followed by one cycle of 72°C for 7 min. The PCR amplified products were checked in 1.5% agarose gels by electrophoresis. After that, the gel was stained with ethidium bromide, washed with distilled water, and then photographed under a UV transilluminator.

### 2.4. Serotyping

*Vp*_AHPND_ strains isolated were serotyped by slide agglutination test using commercial antisera test kit (Denka Seiken Co., Tokyo, Japan) following the instruction of company. The bacteria were subcultured on NA supplemented with 1% NaCl. After that, a loopful of colonies was collected and mixed well in 4 mL of 3% NaCl solution supplemented with 5% glycerol. The mixture was autoclaved at 121 °C for 60 min. and then centrifuged at 30,000 rpm for 20 min. Next, the supernatant was removed and the pellet was dissolved in 500 µl of 3% NaCl solution. This mixture was used to examine for O antigen. The remaining colonies on NA plate was used directly for K antigen examination.

### 2.5. Examination of Pathogenicity of Vp_AHPND_ Strains

It is reported that there is a relation between antibiotic resistance and virulence of bacteria. In this study, five *pirAB^vp^* gene positive strains that showed five different antimicrobial resistance patterns were selected and the pathogenicity of each strain was examined. Pathogenicity of *Vp*_AHPND_ strains were examined while using challenge test by immersion method following protocol described by Loc et al. [7]. Briefly, white leg shrimp at about 1 g was immersed in jars containing 5 L sterilized seawater (20 ppt) with aeration. In challenge treatments, the concentration of *Vp*_AHPND_ in each jar was adjusted at 10^6^ CFU mL^−1^. In control group, no AHPND strains were added into the jars. The experiment was observed for at least one week. Experimental shrimp was observed for cumulative mortality and clinical signs. Moribund shrimp was picked up for re-isolation of *Vp*_AHPND_. Several moribund shrimps were also used for histopathological examination of HP using hemotoxyline and eosin-phloxine (H&E) stain, as described by Lightner [13].

### 2.6. Examination of Antimicrobial Susceptibility

The antimicrobial susceptibility test was done using the disk diffusion method. The procedures were based on the guidelines of the clinical and laboratory standards institute [14]. Ten antimicrobial agents (Becton, Dickinson and Company, Franklin Lakes, NJ, USA) were used in this study, including ampicillin (10 µg), chloramphenicol (30 µg), colistin (10 µg), kanamycin (30 µg), gentamicin (10 µg), nalidixic acid (30 µg), ofloxacin (5 µg), oxytetracycline (30 µg), tebipenem (10 µg), and streptomycin (10 µg). The CLSI criteria were used to interpret the susceptibility results of these antimicrobial agents except for colistin and tebipenem. Criteria for colistin and tebipenem were based on the instruction of Galani et al. [15] and Fujisaki [16], respectively.

## 3. Results

### 3.1. Prevalence of Vp_AHPND_ in Shrimp, Molluscan Shellfish, and Water Samples in the Mekong Delta

Table 1 shows the prevalence of *Vp*_AHPND_ strains in shrimp, molluscan shellfish, and water samples in this study. The *Vp*_AHPND_ strains were isolated in 2 of 298 (0.7%) molluscan shellfish samples from retail shops, 7 of 71 (9.9%) shrimp samples from shrimp ponds and 2 of 42 (4.8%) water samples from shrimp ponds. All *Vp*_AHPND_ strains isolated were *toxR* gene positive (Table 2).

### 3.2. Serotypes

Sixteen *Vp*_AHPND_ strains were serotyped (Table 2). They were classified into two types of O antigen, including O1 and O3, in which O1 was the predominant. Regarding to K antigen, nine strains were typed into several antisera, including K25, K31, K64, K68; whereas, seven strains did not react to any K antisera.

### 3.3. Pathogenicity of Vp_AHPND_

*Vp*_AHPND_ strains examined in challenge experiments showed virulence to white leg shrimp (Table 2). After one week of challenge, the cumulative mortality of shrimp infected *Vp*_AHPND_ strains varied from 40 to 100%. The high mortality occurred in the first three days after infection. No mortality was observed in unchallenged shrimp.

The challenged shrimp showed clinical signs of AHPND, such as atrophied and pale HP, empty stomach, and midgut (Figure 1). The histopathology of HP of the challenged shrimp indicated sloughing and necrosis of HP tubule epithelial cells and hemocytic infiltration surrounding HP tubules (Figure 2).

### 3.4. Antimicrobial Susceptibility

Table 3 showed that most of the *Vp*_AHPND_ strains were resistant to colistin (100%), ampicillin (93.8%), and streptomycin (87.5%).

Of 16 *Vp*_AHPND_ strains examined, one strain showed resistance to five antimicrobial agents, three to four, nine to three, and three to two antimicrobial agents (Table 2). Generally, all (100.0%) strains showed resistance against more than one antimicrobial agent.

## 4. Discussion

*V. parahaemolyticus* naturally distributes in estuarine and marine environment. It is known as one of leading cause of foodborne disease in human and the infection could be via eating raw or undercook seafood [17]. The high prevalence of this bacterium is usually recorded in shellfish species [18,19]. Human pathogenic *V. parahaemolyticus* carries several virulent genes in which thermostable direct hemolysin (*tdh)* and *tdh-* related hemolysin *(trh)* play as the most important factors [20]. The *pir^vp^* genes could be recently acquired by *V. parahaemolyticus* and caused shrimp disease [12]. Several reports indicated that shrimp pathogenic strains did not posse human pathogenic genes and their DNA profiles were distinct from those of human pathogenic strains [21,22]. This study tried to detect *Vp*_AHPND_ strains from environment in the Mekong Delta to know the prevalence of this pathogen in this area after the first outbreak of AHPND in 2010. *V. parahaemolyticus* related to AHPND is usually isolated from shrimp ponds [21,22]. Nevertheless, in this study, this pathogen was isolated not only in shrimp and water from shrimp ponds, but also in molluscan shellfish from retail shops. The shellfish were usually sold immediately after they were harvested at the coast and many retail shops in this study were located near the coast in the Mekong Delta; therefore, the prevalence of pathogenic *V. parahaemolyticus* in retail shellfish seemed to reflect that in the environment in this coastal area. Moreover, we would like to know whether *Vp*_AHPND_ strains that were isolated in this study cause AHPND in shrimp. Several *Vp*_AHPND_ strains isolated were selected and subjected to challenge experiment. It is postulated that there is a relationship between virulent and antibiotic resistance. For instant, the increased virulence and the advent of antibiotic resistance often obtain almost simultaneously [23]. Thus, five AHPND strains showing different antimicrobial resistance patterns were selected for challenge experiment. Notably, the selected strains showed AHPND pathogenicity to experimental white leg shrimp. These results suggest that AHPND *V. parahaemolyticus* seems to present widely in environment in the Mekong Delta. FAO [1] indicated that shrimp can be infected AHPND via infected water and fed infected shrimp. A recent study by Boonyawiwat et al. [24] reported that the risk factors of AHPND occurrence in the Mekong Delta related to farm size, cleaning pond bottom method, using AHPND affected water, water depth, weather change, and hatchery. Further studies should clarify other factors related to the outbreak of AHPND.

The previous study by Kongueng et al. [22] reported *Vp*_AHPND_ strains that were isolated from shrimp ponds in Thailand had a unique O1 antigen with three types of K antigen. However, the following study by Chonsin et al. [21] demonstrated three types of O antigen, including O1, O3, and O8 and three types of K antigen of *Vp*_AHPND_ strains from shrimp ponds in Thailand. The present study detected two types of O antigen comprising of O1 and O3 and five types of K antigen of *Vp*_AHPND_ strains. It was known that the virulent genes of AHPND in shrimp are encoded on plasmid of *V. parahaemolyticus* [8]. Han et al. [12] characterized AHPND plasmid and described *pir^vp^* genes are flanked by a mobile genetic element, which can promote horizontal gene transfer. A variation of serotypes of *Vp*_AHPND_ strains suggested that the virulent genes could be transferred among *V. parahaemolyticus* strains. The existence of *pir^vp^* genes was also reported in non-*V. parahaemolyticus* strains, such as *V. harveyi*-like strains [25] and *V. campbellii* [26]. Factors affecting the transfer of the virulent genes should be investigated.

Most of *Vp*_AHPND_ isolated in this study were resistant to colistin, ampicillin, and streptomycin, and all AHPND strains performed multiple antimicrobial resistances. Obaidat et al. [27] also reported *V. parahaemolyticus* strains isolated from seafood samples from some developing countries showed a high resistance rate to colistin (100%). The high resistance rates to ampicillin and other antimicrobials were also observed on *Vp*_AHPND_ strains in other countries. Kongueng et al. [21] reported that 100% of *Vp*_AHPND_ strains isolated from Thailand were resistant to ampicillin and erythromycin. Han et al. [28] indicated 100% of *Vp*_AHPND_ strains isolated from Mexico were resistant to ampicillin and oxytetracycline. In fact, several antimicrobial agents such as ampicillin and streptomycin has been introduced and used in environment for the last some decades; therefore, the resistance of bacteria to those antimicrobial agents could be facilitated via selective pressure [29,30]. The resistance of bacteria to particular antimicrobial agents because of bacteria’s innate ability [31] or acquiring antimicrobial resistant plasmids [32] was also noted.

## 5. Conclusions

In conclusion, *Vp*_AHPND_ seems to be widely present in environment in the Mekong Delta. The serotype of *Vp*_AHPND_ strains was diversity. Additionally, *Vp*_AHPND_ strains showed resistances to several antimicrobial agents. These finding can be used for understanding the risk of AHPND in shrimp in the Mekong Delta.

## Figures and Tables

**Figure 1 biology-09-00312-f001:**
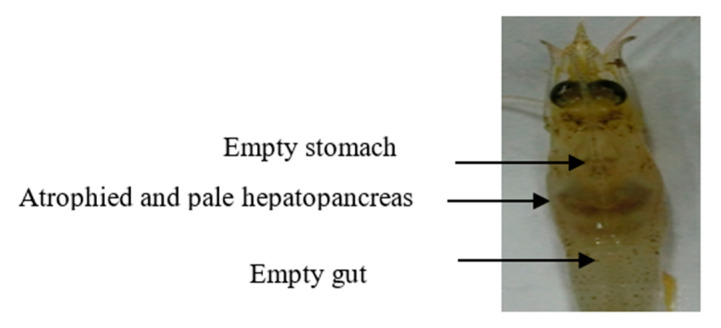
Clinical signs of shrimp challenged with AHPND strains.

**Figure 2 biology-09-00312-f002:**
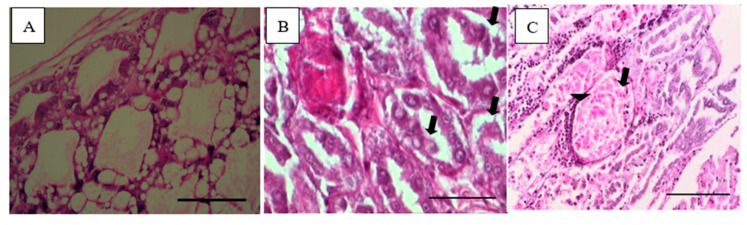
Histopathology of shrimp hepatopancreas (HP). Unchallenged shrimp showed normal structure of shrimp HP (**A**). Challenged shrimp showed sloughing of HP tubule epithelial cells ((**B**), arrows), sloughing and necrosis of tubule epithelial cells ((**C**), arrow) and hemocytic infiltration surrounding HP tubules ((**C**), arrowhead). H&E stain. Magnification 400X. Scale bars = 50 µm.

**Table 1 biology-09-00312-t001:** Prevalence of *Vp*_AHPND_ in shrimp, molluscan shellfish, and water samples in the Mekong Delta, Vietnam.

Origins	Sample Types	No. of Samples	No. of *Vp*_AHPND_ Positive Samples (%)
Seafood samples			
Retail shops	Molluscan shellfish	298	2 (0.7)
	Shrimp	32	0 (0.0)
	Subtotal	330	2 (0.6)
Farms	Molluscan shellfish	16	0 (0.0)
	Shrimp	71	7 (9.9)
	Subtotal	87	7 (8.0)
Total		417	9 (2.2)
Water samples			
Farms	White hard clam farms	22	0 (0.0)
	Shrimp ponds	42	2 (4.8)
Total		64	2 (3.1)

**Table 2 biology-09-00312-t002:** Characterization of acute hepatopancreatic necrosis disease (AHPND) *V. parahaemolyticus* isolated in the Mekong Delta.

No. of Samples	Strain Names	Serotypes	Pathogenic Genes		Species-Specific Gene (*toxR*)	Pathogenicity to Shrimp	Antimicrobial Resistant Patterns	Origins
			pirA^vp^	pirB^vp^				
1	VP-AHPND-1	O1:K64	+	+	+	+	AMP-COL ^c^	Shellfish in shop
2	VP-AHPND-2	O1:K68	+	+	+	+	COL-STM	Shellfish in shop
3	VP-AHPND-3	O1:K_UT_ ^a^	+	+	+	+	AMP- COL- STM	Water at shrimp pond
4	VP-AHPND-4	O1:K_UT_	+	+	+	+	AMP-COL-OTT-STM	Water at shrimp pond
5	VP-AHPND-5	O1:K68	+	+	+	+	AMP-COL- NAL-OTT-STM	Shrimp at shrimp pond
6	VP-AHPND-6	O1:K_UT_	+	+	+	NE ^b^	AMP-COL-OTT-STM	Shrimp at shrimp pond
	VP-AHPND-7	O1:K_UT_	-	+	+	NE	AMP-COL-OTT-STM	Shrimp at shrimp pond
7	VP-AHPND-8	O1:K25	+	+	+	NE	AMP-COL	Shrimp at shrimp pond
8	VP-AHPND-9	O1:K25	+	+	+	NE	AMP-COL-STM	Shrimp at shrimp pond
	VP-AHPND-10	O1:K_UT_	+	+	+	NE	AMP-COL-STM	Shrimp at shrimp pond
9	VP-AHPND-11	O1:K_UT_	+	+	+	NE	AMP-COL-STM	Shrimp at shrimp pond
	VP-AHPND-12	O1:K25	+	+	+	NE	AMP-COL-STM	Shrimp at shrimp pond
10	VP-AHPND-13	O1:K25	+	+	+	NE	AMP-COL-STM	Shrimp at shrimp pond
	VP-AHPND-14	O1:K_UT_	+	+	+	NE	AMP-COL-STM	Shrimp at shrimp pond
11	VP-AHPND-15	O3:K31	+	+	+	NE	AMP-COL-STM	Shrimp at shrimp pond
	VP-AHPND-16	O3:K31	-	+	+	NE	AMP-COL-STM	Shrimp at shrimp pond

^a^: Untypeable; ^b^: Not examined in challenge experiment; ^c^: AMP: Ampicillin; COL: Colistin; NAL: Nalidixic axid; OTT: Oxytetracycline; STM: Streptomycin.

**Table 3 biology-09-00312-t003:** Antimicrobial resistances of *Vp*_AHPND_ strains (*n* = 16).

Antimicrobial Agents	No. of Resistant Strains (%)
Colistin	16 (100.0)
Ampicillin	15 (93.8)
Streptomycin	14 (87.5)
Oxytetracycline	4 (25.0)
Nalidixic acid	1 (6.3)
Chloramphenicol	0 (0.0)
Gentamicin	0 (0.0)
Kanamycin	0 (0.0)
Ofloxacin	0 (0.0)
Tebipenem	0 (0.0)

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
