# Peer review of "Prevalence of *Vibrio parahaemolyticus* Causing Acute Hepatopancreatic Necrosis Disease of Shrimp in Shrimp, Molluscan Shellfish and Water Samples in the Mekong Delta, Vietnam"

_biology, 2020, doi:10.3390/biology9100312_

Round 1

Reviewer 1 Report

The short communication titled "Prevalence of Vibrio parahaemolyticus Causing Acute Hepatopancreatic Necrosis Disease of Shrimp in Shrimp, Molluscan Shellfish and Water Samples in the Mekong Delta, Vietnam" refers occurrence of pathogenic bacteria in food and water samples. The specific strain of V. parahaemolyticus is important due to economic and health reasons.

General comments

In general, the text needs to be checked for formatting end selling. In my opinion, the authors should discuss more risks posed by V.  parahaemolyticus on AHPND in shrimp in the Mekong Delta but also on food safety.

Detailed comments

Line 35: I suppose that it should be "white" instead "while.

Line 83: On what basis was the selection made. Please explain. And there is no information about the unchallenged sample of shrimps.

Line 113: The citing and numbering of the table are incorrect. I suggest combining tables 2 and 4 because three columns are exactly the same.

Line 114: What means "40 to 100%". In the Material and methods, there is no information on how the number of infected shrimps was counted.

Line 119: Fig. 1 shows the histopathological image of shrimp hepatopancreas. The sentence suggests a different kind of images.

Line 142: If there were any signs of illness on molluscan shellfish?

Reviewer 2 Report

The study is well presented but there is need to present a brief summary of the protocol used for detection of pathogenic gene (item 2.3) and serotyping (item 2.4)

Reviewer 3 Report

The authors of the manuscript investigated the prevalence of V. parahaemolyticus, leading cause of seafood-derived food poisoning, having a huge impact on the loss of shrimp as well as on the human health.

The authors investigated seafood- and water samples in Vietnam area, used culture-depended methods, and characterized the isolated strains for their pathogenicity (pirABvp), serotypes and antimicrobial resistance.

In general, the study is very important and provides interesting data of the prevalence of Vibrio strains as well as a better understanding of their potential virulence.

However, I believe that some additional experiments or data (if possible), could support this or further study:

I am wondering, if you could identify toxins (TDH or TRH) in your strains? As described by Broberg et al. (2012), the identification of the tdh gene in samples has shown to be a more accurately predict virulence than serotyping.

Did you perform whole genome sequencing of the isolated strains and compare them to the related outbreaks strains?

And if so, could you please provide phylogenetic analysis (e.g. based on Single Nucleotide Polymorphisms (SNPs) Identification in the genomes)?
